# Clip–Detector Using a Neodymium Magnet to Locate Malignant Tumors during Laparoscopic Surgery

**DOI:** 10.3390/s22145404

**Published:** 2022-07-20

**Authors:** Kicheol Yoon, Kwang Gi Kim, Jun-Won Chung, Won Suk Lee

**Affiliations:** 1Medical Devices R&D Center, Gachon University Gil Medical Center, Incheon 21565, Korea; kcyoon98@gachon.ac.kr; 2Department of Biomedical Engineering, College of Medicine, Gachon University, Incheon 21565, Korea; 3Department of Biomedical Engineering, College of Health Science, Gachon University, Incheon 21936, Korea; 4Department of Health Sciences and Technology, Gachon Advanced Institute for Health Sciences and Technology (GAIHST), Gachon University, Incheon 21565, Korea; 5Department of Gastroenterology, Gachon University Gil Medical Center & College of Medicine, Gachon University, Incheon 21565, Korea; 6CAIMI Co., Ltd., #206 Building of Incheon Startup Park, 204 Convensia-daero (Songdo-dong), Yeonsu-gu, Incheon 22004, Korea; 7Department of Surgery, Gachon University Gil Medical Center & College of Medicine, Gachon University, Incheon 21565, Korea; lws@gilhospital.com

**Keywords:** endoscope, clip–detector, neodymium magnet, tumor location detection, laparoscopic surgery, magnetic coupling

## Abstract

During laparoscopic surgery for colorectal or gastric cancers, locating the tumor for excision is difficult owing to it being obscured by mucous membranes. Therefore, a clip can be installed around the tumor, which can be located using a sensor. Most of the clip–detectors developed thus far can only detect tumors in either the colon or stomach and require a wire to connect the clip and detector. This study designs a clip and detector that can locate a tumor in the stomach and colon. The clip contains a neodymium magnet that generates a magnetic field, and the detector includes a Colpitts oscillator that allows magnetic coupling of the clip and detector. After installing the prepared clip at the tumor location, the detector is used to locate the clip. To test the clip and detector, we conducted animal experiments, during which four clips were installed in the colon and stomach of a mini pig. We succeeded in locating the clips within 2.17 and 3.14 s in the stomach and colon, respectively, which were shorter than the detection times reported in previous studies. The demand for laparoscopic surgery and endoscopes is predicted to increase owing to this method.

## 1. Introduction

Colorectal and stomach cancers have ranked 3–5 globally since 2012 [1]. Owing to the development of preventive diagnostic medicine, endoscopic technology is rapidly developing, and the demands for diagnosis and laparoscopic surgery are increasing simultaneously [2]. Early detection of gastrointestinal (colon/stomach) cancer is possible using endoscopic diagnosis, and a five-year survival rate of over 90% can be guaranteed through biopsy and surgical treatment. Safe surgery involves promptly extracting the tumor by accurately locating it. Furthermore, the purpose of laparoscopic surgery is to quickly identify the location of the tumor and to accurately and safely extract it, thereby shortening the overall operation time. This surgical technique involves a simple operation with minimal side effects and pain, and the patient recovers rapidly; however, determining the location of the mucosal resection to remove the tumor from the gastrointestinal cavity is difficult [2,3,4,5]. More specifically, in laparoscopic gastrointestinal cancer (or colon cancer) surgery, malignant tumors present in the visceral cavity due to mucous membranes cannot be seen in the parietal (or cecum) of the cavity, and to remove the tumor, the location of the tumor present in the visceral (or cyclic folds) must be determined. Therefore, three days before surgery, an endoscope is used to find tumors present in the visceral (or cyclic folds) and a marker (or a drug that can indicate the location of the tumor) is installed around the tumor. Additionally, during surgery, the marker position inside the organ (visceral or circular folds) can be found using a device that can detect the marker outside the organ (parallel or cecum). Since the marker’s location is the location of the tumor, the exact location can be identified by removing the tumor, and this method can be the exact determinant of the incision site for tumor removal. Hence, markers used during laparoscopic surgery play a very important role in surgery. It is important to locate the marker quickly and extracting the tumor quickly at the scheduled surgery (anesthesia) time can be a very important factor directly related to the patient’s life protection. Thus, it is very important to design a high-performance maker, and the reason for this is to propose an idea on how the operator can easily perceive markers when they are found in a chaotic surgical site.

Commonly used methods for locating tumors are fluorescent staining using indocyanine green, ink tattooing, and local autologous labeling [6,7,8,9,10]. These methods have some side effects, although they can determine the location of the tumor [8]. Fluorescent staining using indocyanine green is expensive and causes allergic reactions owing to the presence of iodine. The ink spot method involves a long procedure and causes granuloma formation and peritonitis [7,8]. Additionally, autologous local markers bleed profusely and cause cirrhosis of the liver. Furthermore, autologous blood localization is difficult and time-consuming, which makes this method economically disadvantageous. To overcome these disadvantages, techniques for locating a tumor using a sensor that can detect a clip are being studied [11,12,13,14,15,16]. More specifically, a clip is attached to the tumor site, and research is underway on a method for locating this clip using a sensor, thereby locating the tumor that needs resection. Analyzing the research case, the time to locate the tumor must be shortened. In addition, the sensor should be able to detect tumors in the stomach and large intestine simultaneously and be robust. At this time, the time required to locate the tumor through the sensor should be 5 s.

Radio frequency identification (RFID) [11] and the open–close clip closure method [14] have the advantage of being able to simultaneously detect tumors in the stomach and colon. In addition, the clip and the sensor are integrated to ensure robustness. However, the RFID method [11] requires 40.5 and 38.4 s to detect a tumor in the stomach and colon, respectively, and the open–close clip closure method [14] has a relatively long time as 24.9 and 18.7 s, respectively, for detecting tumors in the stomach and colon.

A sensor using RFID technology can only detect the location of a malignant tumor in the stomach [16]. The time required for this detection is as long as 25 s [12,13,15]. This technology uses a magnet, and a wire is connected between the clip and magnet sensor. However, the wire connection is not sturdy and wobbles. Therefore, the location of the tumor changes as much as the length of the wire, which makes it difficult to determine the exact location of the tumor and may cause the extraction to fail. In addition, only tumors in the colon can be detected, and the time required for tumor detection is 456, 342, and 15–90 s, respectively. The characteristic of [11,12,13,14,15,16] is that the time required to detect a tumor exceeds 5 s; therefore, these technologies are unsuitable for use in the operating room. To overcome these shortcomings, a sensor capable of locating a tumor in the stomach and colon simultaneously and accurately is required.

This study proposes a ferromagnetic clip and clip-detection sensor that can simultaneously locate malignant tumors in the stomach and colon quickly and accurately. The clip is constructed using a ferromagnetic material to ensure sturdiness and miniaturization, such that it can pass through the working channel (diameter: 4 mm) of a gastroscope and trocar. The introduction presents the motivation for the research, Section 2 and Section 3 describe the design methods and experimental results, and Section 4 and Section 5 present the discussion and conclusion.

## 2. Analysis of the Magnetic Coupling

In laparoscopic surgery, malignant tumors present in the colon (or stomach) interior (circular folds or visceral) are not visible outside (cecum or parietal) of the colon, so it is not easy to locate the tumor in the cavity [4]. Therefore, to locate the tumor on the inner wall of the stomach (circular folds or visceral), the tumor is located through an endoscope three days before surgery (flow chart on the left), as shown in Figure 1, and then the clip is pre-installed around the tumor. Then, during laparoscopic surgery, the trocar is docked to locate the detector (picture on the right) and the clip present inside the colon (or stomach) in the cavity.

If the detector finds the location of the clip, an alarm (picture on the right) is generated in the detector to guide the location of the clip. Therefore, the tumor can be safely removed, and the surgery is over. In the process of finding a clip using a detector, sine waves generated in the oscillation are transferred to the detector, as shown in Figure 2. A magnet sensor is attached to the end of the detector.

At this time, if the current for the sine wave generated from the oscillator reaches the sensor, the current is generated in the sensor and the magnetic flux density (*B*) is formed as in Equation (1). Assuming that the permeability is constant, if the magnetic flux density (*B*_1_) of the sensor is generated, a magnetic field (*H*_1_) is formed as in Equation (2) [15,16,17,18,19].
(1)$$B_{1,2}=\mu_{0}H_{1,2}\;\left[\mathrm{Nm}/A\right]$$
(2)$$H_{1,2}=\frac{B_{1,2}}{\mu_{0}}\;\left[T\right]$$
where the *B* and *µ*_0_ are magnetic flux density and permeability of free space (4*π* × 10^−7^ H/m). Therefore, the energy (*H*_1_ and *ω*_1_) generated from the magnet sensor passes through the mucous membrane of the colon (or stomach). After that, energy is transmitted (TX) to the neodymium magnet (NeFe35) present inside the clip as shown in Figure 3, where *ω* is angular frequency which is *2πf* and the *f* is frequency.

A neodymium magnet is a magnet with the strongest magnetic force among permanent magnets. Neodymium magnets have excellent machinability of various shapes. Therefore, the neodymium magnet is made of iron and boron, and it has the maximum magnetic energy among permanent magnets. Due to the ferromagnetic properties of the neodymium magnet, a strong magnetic flux density (*B*_2_) is generated as in Equation (1), and the magnetic field (*H*_2_) increases. Therefore, as in Equation (3), the neodymium magnet operates as a magnet by generating a strong magnetic force (*F*). At this time, *D* means the size of the neodymium magnet, and as the size of the magnet increases, the magnetic flux density (*B*_2_) increases. Thus, the magnetic field *H*_2_ will increase, and the magnetic force *F* will increase. Additionally, a magnetic field (*H*_2_) is generated along with the induction frequency (*ω*_2_) [15].
(3)$$F=\frac{\mu_{0}H^{2}D}{2}=\frac{B^{2}D}{2\mu_{0}}\;\left[N\right]$$

As a result, *ω*_1_ and *H*_1_ generated from the sensor are coupled to *ω*_2_ and *H*_2_ generated from the neodymium magnet to form *H*_1_ and *H*_2_ as shown in Equation (4), and *ω*_1_ and *ω*_2_ become *ω*_0_ as shown in Equation (5). Therefore, signals are exchanged (TX/RX) between the sensor and the clip (neodymium magnet) [17].
(4)$$H_{0}={\left(\frac{H_{2}}{H_{1}}\right)}^{2}\;\left[T\right]$$
(5)$$\omega_{0}=\sqrt{\omega_{1}\omega_{2}}\;@\;\omega =2\pi f$$

Therefore, when the detector detects a clip, *H*_0_ and *ω*_0_ are converted into pulse waves through the Schmitt trigger circuit, and the pulse waves are converted into direct current (DC) form that can get sound from the speaker through the regulator. Therefore, DC signal is transmitted to the speaker and in order to provide clean sound quality without noise, we can convert it into pulse by applying the Schmitt trigger circuit. Therefore, it is possible to increase the accurate search efficiency for the position of the clip. However, an important fact is the analysis of the distance between the magnet and the sensor. In the difference between the magnet and the sensor, the coupling frequency (*f_o_*) is the same, but the magnetic flux density (*B*) and the magnetic field (*H*_0_) are changed. From the simulation results, the magnet and sensor are in inverse proportion to each other. As shown in Figure 4, if the distance between the magnet and sensor is increased, *H* and *B* are decreased, and, therefore, *F* is weakened as given by Equation (6) [18,19]. Therefore, as the distance (*d*) between the sensor and the clip increases, the magnetic field (*H*) and magnetic flux density (*B*) decrease, so that the magnetic force (*F*) decreases as the square of the distance [15,16,17].
(6)$$F=\frac{BH}{2\pi d^{2}}\;\left[N\right]\;$$

In contrast, as the distance between the magnet and sensor is decreased, *H* and *B* are shortened, and therefore *F* becomes strong. Additionally, to increase *H*, the size of the magnet must be increased. Therefore, the size of the magnet must also be considered when determining *H* that is sufficient to pass through tissues. The inner diameter of the magnet in this study is determined to be 1.5 mm (*d_n_*), height (*h_n_*) is 3.4 mm, thickness (*t_in_*) is 0.5 mm, and total diameter (D) is 2.5 mm. Additionally, the diameter (*t_c_*) and height (*h_c_*) of the clip are 2.7 mm, respectively, and the working channels of the detector are 2.8 mm (stomach) and 3.2 mm (colon), respectively.

Considering the thickness of the mucosa between the magnet present in the clip and sensor (t_m_) (top: 2 cm) and the cavity distance (d) between the mucosa and sensor, *F* is calculated as 838.4, as shown in Figure 5, measured in N. Additionally, *H* and *B* are calculated as 1.035 A/m and 1.33 T, respectively, and the polarization direction is changed by 0–10°.

## 3. Design and Fabrication of the Clip–Detector

Figure 6 shows the schematic circuit and fabricated printed circuit board containing the clip–detector. As shown in the figure, the circuit consists of a Colpitts oscillator, amplifier, Schmitt trigger, regulator, relay switch, and speaker.

In particular, the Colpitts oscillator is composed of parallel capacities (*C_x_*: *C*_1_//*C*_2_, see Figure 6) as shown in Equation (7), and thus the quality factor (Q) is increased due to high *C_x_*. This high Q is a decisive factor in accurately locating the clip. Therefore, the oscillation frequency *ω_r_* is determined by the adjustment of the *C_x_* value as shown in Equation (8). However, the inductance *L_x_* may act as a resistor *R_x_* as shown in Figure 6.
(7)$$C_{x}=\frac{C_{1}C_{2}}{C_{1}+C_{2}}$$
(8)$$\omega_{r}=\frac{1}{\sqrt{C_{x}L_{x}}}\;@\;{L_{x}}_{\;}=R_{x},\;\omega_{r}=2\pi f$$

The Colpitts oscillator is connected to a crystal and the frequency of the crystal is set to operate from 1.56 to 1.57 kHz. Therefore, the oscillator generates a sine wave with a frequency of 1.56–1.57 kHz, and this sine wave is transmitted to the sensor of the detector through TX of the relay switch (S/W).

The sine wave (*H*_1_ and *ω*_1_) output from the sensor of the detector is transmitted to the clip, and the signals (*H*_2_ and *ω*_2_) generated from the clip are combined with the signal generated from the sensor to generate the coupling signal (*H*_0_ and *ω*_0_). Therefore, the sensor and clip exchange signal with each other. The signal detected in the clip is transferred to the RX of the relay switch and converted into a pulse wave through the Schmitt trigger circuit, improving the sound quality of the detected signal and suppressing the noise. The pulse wave provides a function so that sound can be heard from the speaker by converting it into direct current (DC) form through the regulator. Therefore, the speaker notifies the fact that the clip has been detected by generating an alarm.

During the PCB fabrication process, the fabrication of the substrate used a Fr4 substrate with a dielectric constant of 4.6 and a thickness of 3.2 T. The magnet is used for combination of the nickel and gold, and the material of the coating with the clip is used by parylene C. In addition, the detector is used for 3D printing technique which is shown in Figure 7.

## 4. Results and Discussion

### 4.1. Circuit Simulation Results

The oscillations produced by the designed Colpitts oscillator are shown in Figure 8. The oscillation jogging is divided from oscillation start-up to oscillation steady-state oscillation. Herein, the start-up range is from 2.0 to 10 ns and the steady-state ranges from 10 to 30 ns. Therefore, the operating range voltage at start-up is 0–2 V, and that of the steady-state oscillation is maintained at 2 V. The steady-state oscillation, therefore, becomes a stable frequency source that is supplied to the magnet in the clip.

Figure 9 shows the result of simulating the pulse signal of the Colpitts oscillator. From the figure, the bias and output voltages of the oscillator are 1.5 and 2.0 V, respectively. More specifically, 2.0 V is obtained between 10 and 30 ns, which is the steady-state condition (see Figure 9). Additionally, the oscillation frequency (*f*_1_) is 1.57 kHz and duty cycle is 50%.

If the detector containing the oscillator detects the magnet present in the clip, the frequency (*f*_2_) shifts to 1.595 kHz and the sensing voltage is 2.03 V, as shown in Figure 10. Therefore, since the detector and clip are magnetically coupled, the frequency (*f*_0_) and voltage between the detector and clip are 1.595 kHz and 3.33 V, respectively.

### 4.2. Animal Testing and Measurement Results

To test the performance of the designed clip–detector, we conducted animal experiments on a male mini pig (farm) that weighed 40 kgs. The animal was supplied by the experimental animal center of KNOTUS (Songdo Research Center, Incheon, Republic of Korea). We sent an animal institutional review board (IRB) permission request to the animal ethics commission (KNOTUS–IACUC–20–KE617). For this experiment, four clips were installed in the visceral cavity of the mini pig, i.e., in the middle circular muscular layer of the stomach and cecum: sigmoid colon using a gastroscope or colonoscope, as shown in Figure 11.

As shown in Figure 12, we confirmed whether the clips were installed correctly through the endoscope using an external monitor and prepared for laparoscopic surgery.

After installing the clips using an endoscope, we tested the detection performance of the sensor during laparoscopic surgery, as shown in Figure 13a. The detector is inserted into the abdominal cavity using a trocar, and the detector begins to detect the clips in the visceral cavity, as shown in Figure 13b. When the clips present in the stomach (visceral cavity) and colon (circular folds) are detected, the detector produces an amplitude of 3.20 V (at 1.59 kHz), as shown in Figure 13c, and an alarm is generated, which is audible through the speaker. When the detector encounters the clip, the detector is coupled to the clip; therefore, the detection time is 1.11 and 1.18 s for the clip present in the stomach and colon, respectively (investigation time is 2.17 and 3.41 s, respectively). Herein, 1.08 and 1.16 s are the reaction speeds when the detector is coupled with the clip for the stomach and colon, respectively, and the time required to locate the clip in the stomach and colon is 2.17 and 3.41 s, respectively. Additionally, no time delays occur, and the response time for detecting the tumor is short.

The thickness of the mucosa in the stomach and large intestine between the clip and detector is usually 2 and 1 cm, respectively. The detector is separated by a distance of 0.5 cm from the parietal peritoneum above. Additionally, in the colon, the detector was approximately 0.5 cm away from the cecum. Therefore, the distance at which the detector detects the clip is at a level of 2.5 cm (top) and 1.5 cm (colon).

During the animal experiment, the surgeon attempts to locate the clip using the detector without knowing the position of the clip. When the designed detector is docked using the trocar and inserted into the cavity, the time required for the clip–detector to locate the tumor in the stomach and colon is recorded as 2.17 and 3.41 s, respectively.

In addition, the magnet for locating the tumor is inserted into the clip, as reported in [11,14,16], and it is designed to be integrated. This allows the detection module to be wirelessly connected to the clip. Therefore, the errors are minimized when the detector locates the clip. More specifically, if connected with a wire, the error in locating the clip is the maximum of the length of the wire [12,13,15]. However, since the designed clip is connected wirelessly, the error in the position of the clip is reduced. In addition, the proposed clip–detector can locate tumors in the stomach and colon as in [11,14]. That is, refs. [11,14] have the greatest advantage in that they can find both the clip positions for the stomach and colon at the same time. However, the clip–detector can locate the tumor accurately and quickly.

Table 1 compares the time required by the detector designed in this study to locate a tumor and those reported in previous studies. The designed detector can detect a tumor 0.08 times (stomach) and 0.12 times (colon) faster than [14]. Therefore, it can be seen that the designed clip–detector is better than [14].

## 5. Conclusions

This study presents a method for locating a malignant tumor in the stomach and colon using a sensor in the cavity during laparoscopic cancer surgery. Determining the extent of excision for removing gastric and colonic tumors is difficult. Therefore, after installing the clip around the tumor, locating the clip using a detector and extracting the tumor is a feasible solution.

Since the proposed clip–detector employs magnetic field coupling based on a neodymium magnet (clip) detection technology, tissue penetration is easy. Additionally, the amplitude of magnetic coupling has a relatively high gain at 3.33 V and 1.59 kHz. Therefore, if the detector locates the clip installed around the tumor, an alarm is audible from the speaker at an amplitude of 3.20 V (frequency: 1.59 kHz). Therefore, the tumor can be located quickly using the alarm in the operating room.

The designed clip–detector does not have any side effects, unlike the conventional ink tattoo, indocyanine green fluorescence staining, and autologous blood maker methods, and it can quickly locate the tumor, thereby reducing the burden between doctors and patients. Because the clip–detector uses ferrite and a coil, it is inexpensive and easy to manufacture. Additionally, the clip–detector can be mass produced owing to low unit price. Since the clip–detector requires an endoscope, the demand for endoscopes and laparoscopic surgery is expected to increase in the future.

## Figures and Tables

**Figure 1 sensors-22-05404-f001:**
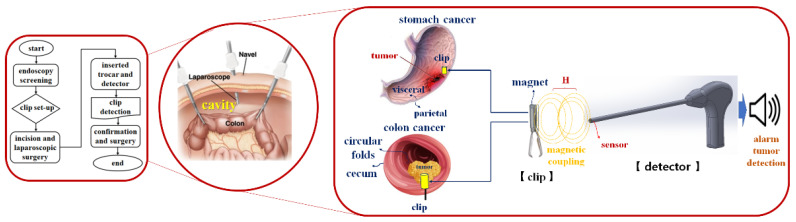
Process of tracking with tumor localization using the clip–detector.

**Figure 2 sensors-22-05404-f002:**
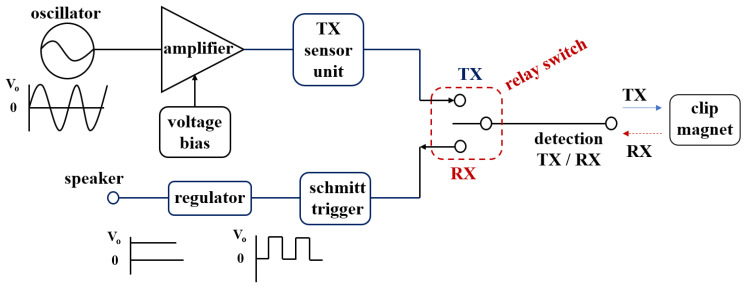
Block diagram of the clip–detector.

**Figure 3 sensors-22-05404-f003:**
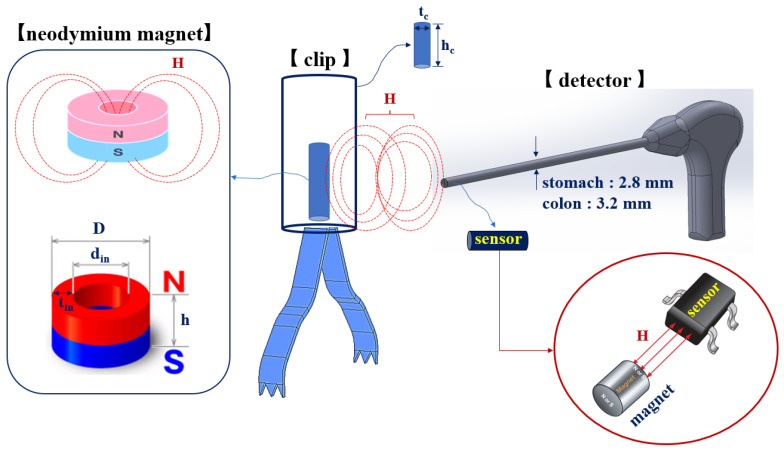
Schematic of the magnetic coupling between the sensor and magnet.

**Figure 4 sensors-22-05404-f004:**
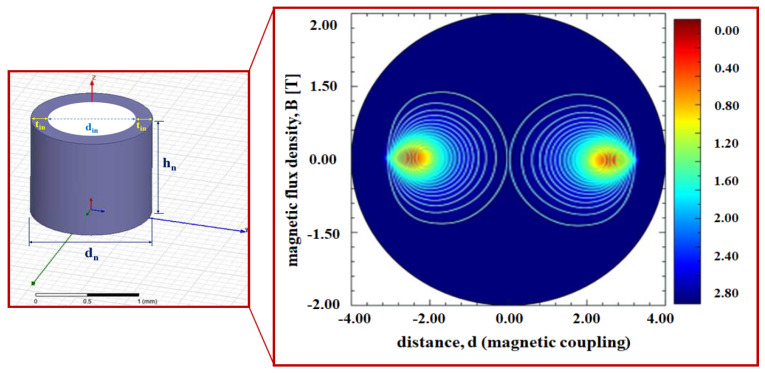
Structure of the magnet and simulation result of the magnetic flux density with respect to the distance from the magnet.

**Figure 5 sensors-22-05404-f005:**
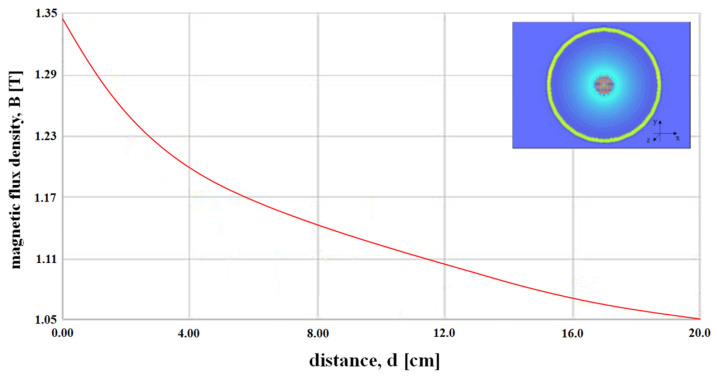
Simulation result of the magnetic flux density with respect to the distance from the magnet.

**Figure 6 sensors-22-05404-f006:**
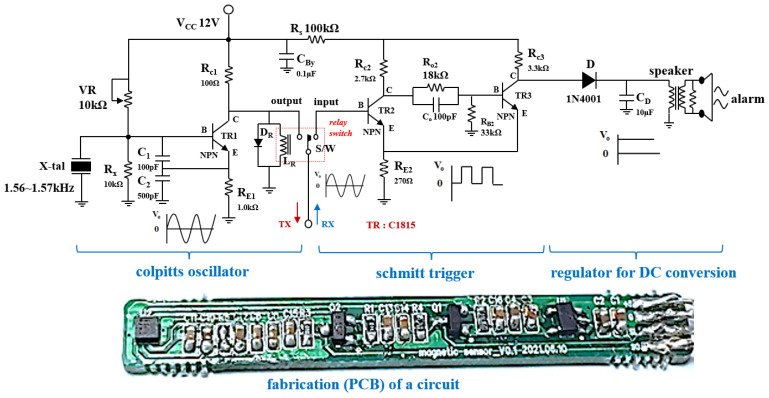
Schematic circuit of the Colpitts oscillator.

**Figure 7 sensors-22-05404-f007:**
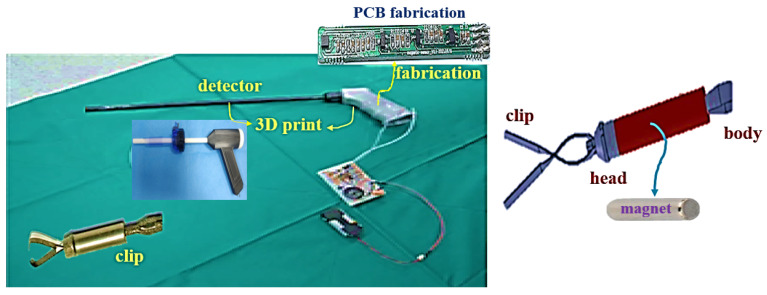
Fabrication of the clip and detector.

**Figure 8 sensors-22-05404-f008:**
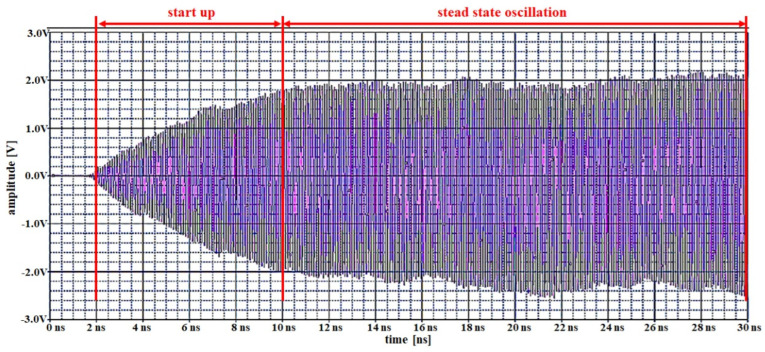
Result of simulating the oscillations of a Colpitts oscillator.

**Figure 9 sensors-22-05404-f009:**
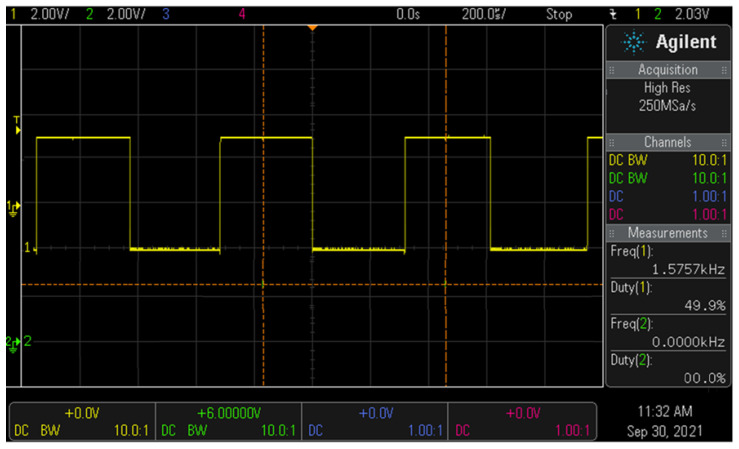
Result of simulating the pulse signal of the Colpitts oscillator.

**Figure 10 sensors-22-05404-f010:**
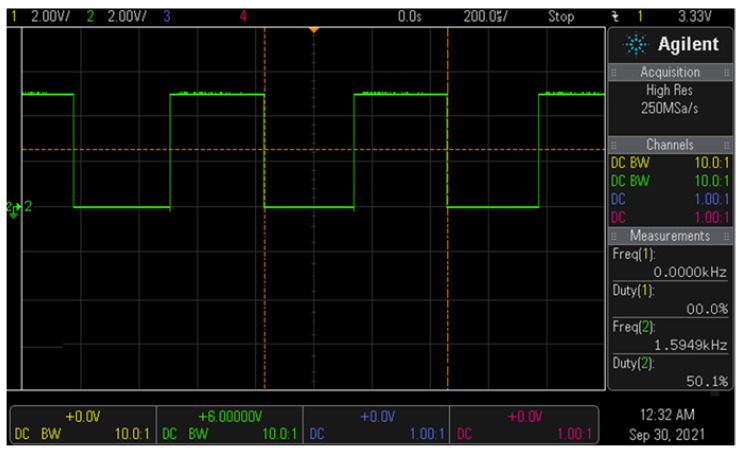
Result of simulating the waveform associated with detecting the clip.

**Figure 11 sensors-22-05404-f011:**
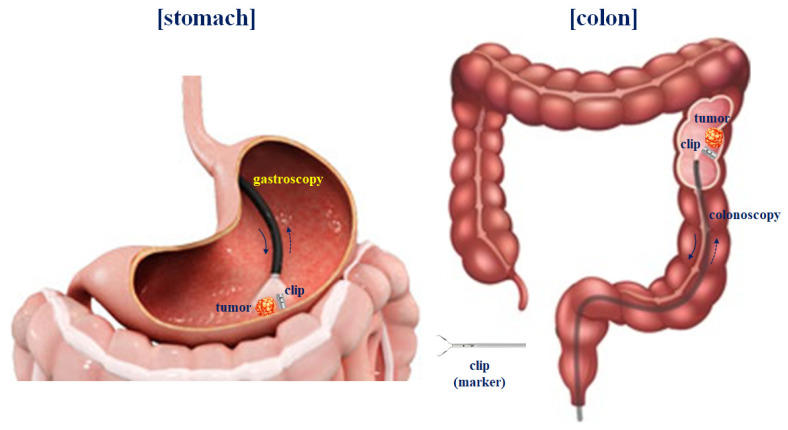
Process of installing clips in the mucosa of the stomach and colon using an endoscope.

**Figure 12 sensors-22-05404-f012:**
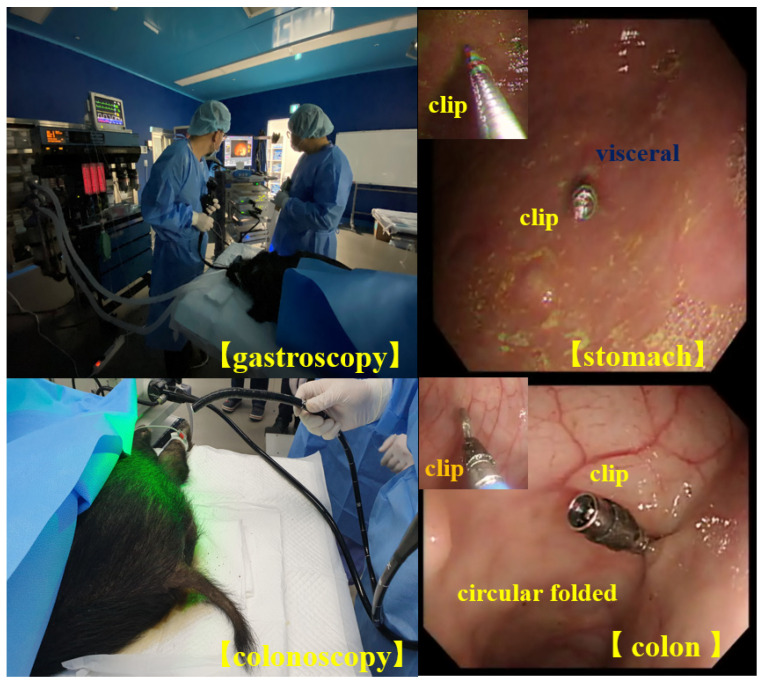
Clips installed in the mucosa of the colon and stomach using guided gastroscopy and colonoscopy.

**Figure 13 sensors-22-05404-f013:**
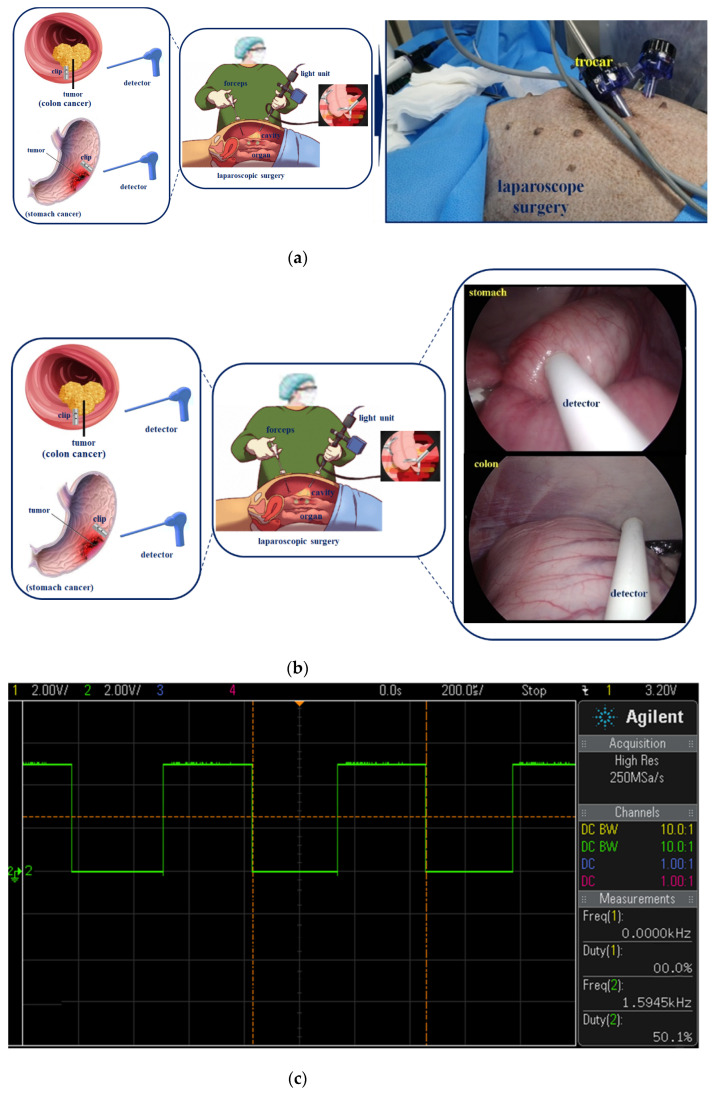
Animal testing of the clip–detector. (**a**) Laparoscope surgery, (**b**) clip detection Appendix A, (**c**) clip detection signal.

**Table 1 sensors-22-05404-t001:** Comparison of the time required to locate a clip using the designed clip–detector and using other methods reported in the literature.

Ref [#]	Time Required to Detect a Clip (s)		Methods
	Stomach	Colon	
This study	2.17	3.41	Neodymium
[11]	40.5	38.4	RFID
[12]	−	15 to 90	Magnet
[13]	−	456	Magnet
[14]	24.9	18.7	Open–close clip closure method
[15]	−	342	Magnet
[16]	25.0	−	RFID

# is the reference number and ref. is the abbreviated version of “reference”.

## Data Availability

The data presented in this study are available upon request from the corresponding author. The data are not publicly available because of privacy and ethical restrictions.

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
