# Peer review of "Clip–Detector Using a Neodymium Magnet to Locate Malignant Tumors during Laparoscopic Surgery"

_sensors, 2022, doi:10.3390/s22145404_

Round 1
Reviewer 1 Report
This paper is on the development of a clip-detecting device that can be used for the location of a tumor during laparoscopic surgery in more accurate and prompt manner than conventional methods. Overall, the story flow in the introduction was nice and also the motivation along with the concept or strategy presented by the authors to locate a tumor appears to be technically sound and feasible.
However, the current manuscript was written mostly with ideas, and it is filled with just unripe results as most of figures consist of conceptual diagrams rather than convincing experimental data. To be more specific, similar stories and conceptual diagrams are repeated too often, and the circuit test and in vivo demo results do not convince this reviewer.
Thus, although the proposed strategy and device concept is novel and looks feasible, it is necessary to substantially revise the manuscript in a way that they substitute or reinforce related experimental data.
1. First, the explanation of the basic idea is too unnecessarily repeated in the paper. For example, the authors’ explanation on the principle of the clip-detector and such a device feature that the clip includes a strong magnet, are repeated in several locations of the manuscript.
2. If a tumor location is already identified, why should such a clip be installed? I mean that it is necessary for the authors to add some more direct and crucial remarks (as to why such a clip needs to be installed when a tumor is found through a screening) from the point of view of the relevant clinical procedure so that readers better appreciate related clinical demands.
3. I am not sure whether it is pertinent to use the expression “wireless communication” in Lines 120-128 because the clip cannot generate any signal actively because it is a passive element. It is necessary to find other expression in terms of the English accuracy.
4. Descriptions or definitions for few parameters in Equation (1) are not provided in the main text. So, provide them. Also, re-write Lines 142 and 143 in terms of its clearness, and check your descriptions for Lines 145-146.
5. It is questionable whether the circuit diagrams presented in Figures 8 and 9 are really appropriate circuits for the implementation of the intended device according to the proposed concept. Also, which part (i.e., sub module) of Figure 8 corresponds to the Colpitts oscillator circuit diagram presented in Figure 9? It is necessary to provide more detailed explanations on this or update the currrent circuit diagrams so that related feature or intended operations be better represented. Also, check the names of each submodule in terms of consistency and amend appropriately if necessary.
6. Was the Colpitts oscillator circuit presented in Figure 9 included only for the generation of an alarm sound to inform related operators that a clip has been detected? Or it functions for other purpose?
7. Is the PCB presented in Figure 9 an implementation of the circuit diagram of the Colpitts oscillator only or the entire circuit presented in Figure 8? Also, it is necessary to check the English phrase “PCB fabricated on a Colpitts oscillator” included in Figure 9 whether it is pertinent.
8. As far as this reviewer knows, Colpitts oscillator is usually used to generate high frequency oscillation signals. However, the frequencies showcased in the result section were distributed only around 1.57-1.59 kHz Also, this reviewer does not understand why the authors employed that mechanism rather than applying other types of oscillators. It is necessary the authors to provide related reasons.
9. Overall, the information described in Lines 170-183 appears to be somewhat trivial rather than providing more important design points or key operation principles.
10. Above all, although the authors presented Figures 12, 13, and 16 as the signals generated when a clip is detected, they appear to be an ordinary square wave that can be easily created using a common function generator. Also, this reviewer does not understand why they are not in a sinusoidal but in a square waveform because the raw signals generated by a Colpitts oscillator are usually sinusoidal. How can this reviewer be convinced that the waveforms presented in the mentioned figures are relevant clip detection signals that can be expected from the implemented device? In this regard, it will be very nice and striking if the authors could provide any demo video that shows a related clip installation and detection procedure using the implemented device (including an alarm sound).
Author Response
Comment 0 :
This paper is on the development of a clip-detecting device that can be used for the location of a tumor during laparoscopic surgery in more accurate and prompt manner than conventional methods. Overall, the story flow in the introduction was nice and also the motivation along with the concept or strategy presented by the authors to locate a tumor appears to be technically sound and feasible.
However, the current manuscript was written mostly with ideas, and it is filled with just unripe results as most of figures consist of conceptual diagrams rather than convincing experimental data. To be more specific, similar stories and conceptual diagrams are repeated too often, and the circuit test and in vivo demo results do not convince this reviewer.
Thus, although the proposed strategy and device concept is novel and looks feasible, it is necessary to substantially revise the manuscript in a way that they substitute or reinforce related experimental data.
Answer 0 :
Every effort has been made to faithfully respond to your comments. Thank you
Comment 1 :
First, the explanation of the basic idea is too unnecessarily repeated in the paper. For example, the authors’ explanation on the principle of the clip-detector and such a device feature that the clip includes a strong magnet, are repeated in several locations of the manuscript.
Answer 1 :
After checking again, I agree. Therefore, I revised all chapters 2-3 and rearranged the duplicate pictures and sentences. Thank you very much for the advice. I feel improved thanks to this.
Comment 2 :
If a tumor location is already identified, why should such a clip be installed? I mean that it is necessary for the authors to add some more direct and crucial remarks (as to why such a clip needs to be installed when a tumor is found through a screening) from the point of view of the relevant clinical procedure so that readers better appreciate related clinical demands.
Answer 2 :
During laparoscopic surgery, tumors inside the organ (stomach or colon) are not visible from the outside. So, when performing laparoscopic surgery, when the abdominal wall is opened, it is not possible to know where the tumor is inside the organ from outside the organ. Therefore, 3 days before surgery, a clip is installed around the tumor through an endoscope. And during surgery, the position of the clip inside the organ is found using the sensor from the outside of the organ. Since the position of the clip is the position of the tumor, the exact position can be confirmed when the tumor is removed. Therefore, it is possible to accurately determine the incision site for extraction. Therefore, during laparoscopic surgery, the clip is used very importantly in surgery. It is important to find the position of the clip quickly, and it is very important to extract the tumor quickly at the scheduled operation (anesthesia) time. Therefore, it is very important to study the clip design with excellent performance. The operating scene is chaotic. Therefore, when a clip is detected, it is very helpful to listen to the alarm sound rather than to observe the waveform during the operation.
Refer to Lines 97-110.
Comment 3 :
I am not sure whether it is pertinent to use the expression “wireless communication” in Lines 120-128 because the clip cannot generate any signal actively because it is a passive element. It is necessary to find other expression in terms of the English accuracy.
Answer 3 :
I revised the “magnetic coupling and signal (energy) exchange” instead of wireless communication. Please refer to lines of 141 and 143 (red color).
Comment 4 :
Descriptions or definitions for few parameters in Equation (1) are not provided in the main text. So, provide them. Also, re-write Lines 142 and 143 in terms of its clearness, and check your descriptions for Lines 145-146.
Answer 4 :
Thanks for the advice. The definitions of expressions have been summarized.
Please refer to 117, 119, 123-124, 126-127, 131-132, 134-137, 139-142 (green).
Comment 5 :
It is questionable whether the circuit diagrams presented in Figures 8 and 9 are really appropriate circuits for the implementation of the intended device according to the proposed concept. Also, which part (i.e., sub module) of Figure 8 corresponds to the Colpitts oscillator circuit diagram presented in Figure 9? It is necessary to provide more detailed explanations on this or update the currrent circuit diagrams so that related feature or intended operations be better represented. Also, check the names of each submodule in terms of consistency and amend appropriately if necessary.
Answer 5 :
The circuit has been completely overhauled. We apologize for causing misunderstanding by briefly expressing only the necessary parts for better understanding. I fixed the improved circuit thanks to your advice. Thanks for the advice. Please refer to Figures 2 and 6. Also refer to the sentences (108-153 and 180-195).
Comment 6 :
Was the Colpitts oscillator circuit presented in Figure 9 included only for the generation of an alarm sound to inform related operators that a clip has been detected? Or it functions for other purpose?
Answer 6 :
Your question is correct. The purpose of this study is to analyze the performance and effect of clips using neodymimum magnets to quickly find the position of the clip installed on the colon or stomach. Therefore, a sensor is needed to detect the clip and an oscillator is needed to operate the sensor. Therefore, we designed a digital oscillator to provide accurate signal detection and clear sound quality for the detection signal. In addition, we constructed a circuit that provides an alarm sound to quickly identify clip detection at the surgical site. Please refer to 97-110 or 108-154 for details.
Comment 7 :
Is the PCB presented in Figure 9 an implementation of the circuit diagram of the Colpitts oscillator only or the entire circuit presented in Figure 8? Also, it is necessary to check the English phrase “PCB fabricated on a Colpitts oscillator” included in Figure 9 whether it is pertinent.
Answer 7 :
The circuit has been completely overhauled. In order to help understanding, only the necessary parts are briefly expressed, leaving the possibility of misunderstanding. Therefore, the circuit has been completely modified. Thanks for the advice. Please refer to Figures 2 and 6.
Also I modified it with fabrication of a circuit (Fig 6).
Comment 8 :
As far as this reviewer knows, Colpitts oscillator is usually used to generate high frequency oscillation signals. However, the frequencies showcased in the result section were distributed only around 1.57-1.59 kHz Also, this reviewer does not understand why the authors employed that mechanism rather than applying other types of oscillators. It is necessary the authors to provide related reasons.
Answer 8 :
I understand the intent of the question. By expressing only important pictures, there was a possibility of misunderstanding. We designed colpitts oscillator, Schmitt trigger to provide digital waveform, and regulator to generate alarm from speaker, and designed relay switch to clarify signal transmission and receiving path. Please refer to Figures 2 and 6.
Most colpitts oscillators are used to generate high-frequency signals. However, this clip detector set the frequency in the range of 1.57-1.59kHz to match the frequency generated by the neodymimum magnet to detect the clip of the stomach (or colon). Thank you.
Comment 9 :
Overall, the information described in Lines 170-183 appears to be somewhat trivial rather than providing more important design points or key operation principles.
Answer 9 :
Added sentences and made improvements. Please refer to 111-154. Also for the circuit, I added more details. See also 180-195.
Comment 10 :
Above all, although the authors presented Figures 12, 13, and 16 as the signals generated when a clip is detected, they appear to be an ordinary square wave that can be easily created using a common function generator. Also, this reviewer does not understand why they are not in a sinusoidal but in a square waveform because the raw signals generated by a Colpitts oscillator are usually sinusoidal. How can this reviewer be convinced that the waveforms presented in the mentioned figures are relevant clip detection signals that can be expected from the implemented device? In this regard, it will be very nice and striking if the authors could provide any demo video that shows a related clip installation and detection procedure using the implemented device (including an alarm sound).
Answer 10 :
I fully understand the intent of the question. Figures 2 and 6 have been improved. We designed colpitts oscillator, Schmitt trigger to provide digital waveform, and regulator to generate alarm from speaker, and designed relay switch to clarify signal transmission and receiving path.
Also, the operation process was submitted to the up-load site for video capture (or supplementary – video abstract).
If a clip is detected in the video, an alarm is generated from the speaker. Please refer to the video. Thank you.

Reviewer 2 Report
In this manuscript, the authors report a ferromagnetic clip and clip-detection sensor to locate tumors in stomach and colon quickly and accurately. The clip material choice, circuit design, and the oscillation pattern are described thoroughly. Then the platform is applied for animal testing. It is confirmed that the time required to locate a clip based on this proposed platform is cut down to 2-3 seconds in stomach and colon. Which shows the promising potential of this platform in clinical applications. Overall, the designs and results are explained in details and conclusions are well supported by the results. Thus, I suggest minor revision. Below are my comments:
Figure 9, the photograph of PCB board is low resolution, please replace with a better quality one. Same problem in Figure 11.
From Figure 13, the authors commented that the detection of clip is based on the frequency shift. Can the authors comment how this frequency shift and amplitude changes with the relative distance between detector and clip? I assume by changing the distance between detector and clip, the frequency shift will also change? Based on this info, the surgeon can tell if detector is moving towards or away from the clip.
Author Response
Comment 0 :
In this manuscript, the authors report a ferromagnetic clip and clip-detection sensor to locate tumors in stomach and colon quickly and accurately. The clip material choice, circuit design, and the oscillation pattern are described thoroughly. Then the platform is applied for animal testing. It is confirmed that the time required to locate a clip based on this proposed platform is cut down to 2-3 seconds in stomach and colon. Which shows the promising potential of this platform in clinical applications. Overall, the designs and results are explained in details and conclusions are well supported by the results. Thus, I suggest minor revision. Below are my comments.
Answer 0 :
Every effort has been made to faithfully respond to your comments. Thank you
Comment 1 :
Figure 9, the photograph of PCB board is low resolution, please replace with a better quality one. Same problem in Figure 11.
Answer 1 :
I increased the resolution of Figure 6 and Figure 8. Thank you
Comment 2 :
From Figure 13, the authors commented that the detection of clip is based on the frequency shift. Can the authors comment how this frequency shift and amplitude changes with the relative distance between detector and clip? I assume by changing the distance between detector and clip, the frequency shift will also change? Based on this info, the surgeon can tell if detector is moving towards or away from the clip.
Answer 2 :
When the Clip and the sensor are coupled, f1 and f2 change to f0 (Equation 5). However, as the distance between the clip and coupling increases, the magnetic flux density (B) is reduced by Eqs. (1), (2), and the magnetic field (H) is also reduced (Eqs. 1, 2). Therefore, when the distance between the sensor and the clip is reduced, the magnetic force (F) is reduced as the square of the distance as in equation (6). Please refer to lines of 156-162 (green).

Round 2
Reviewer 1 Report
In the revised manuscript, the authors presented more convincing data and provided more pertinent figures and circuit diagrams, which dispels all my previous concerns. Especially, the demo video that the authors newly submitted was very striking and also a crucial one that made this reviewer believe the normal operation of the proposed device concept as expected.
However, the explanation that the authors gave in the response letter as an answer to my question seems far better and more sense than what is written in Lines 51–54 of the revised manuscript. To be more specific, it was crucial that the authors pointed out in the response letter that there is usually a time interval (e.g., 3 days) between a tumor detection and corresponding laparoscopic surgical removal, which however was not mentioned in the main text, and that the tumor is invisible from the parietal peritoneum.
Thus, if the authors could revise that part once more so that the reason why such a clip must be installed could appear more clearly, this reviewer believes that their device concept will be more persuasive and attractive to other researchers. In addition, it was my overall feeling that the level of narrative ability for related contents still looks rudimentary, especially for the sections 2 and 3.
Author Response
Comment 0 :
In the revised manuscript, the authors presented more convincing data and provided more pertinent figures and circuit diagrams, which dispels all my previous concerns. Especially, the demo video that the authors newly submitted was very striking and also a crucial one that made this reviewer believe the normal operation of the proposed device concept as expected.
Answer 0 :
Thank you for the compliment and support. We have done our best to reflect your comments. Thanks for the advice.
Comment 1 :
However, the explanation that the authors gave in the response letter as an answer to my question seems far better and more sense than what is written in Lines 51–54 of the revised manuscript. To be more specific, it was crucial that the authors pointed out in the response letter that there is usually a time interval (e.g., 3 days) between a tumor detection and corresponding laparoscopic surgical removal, which however was not mentioned in the main text, and that the tumor is invisible from the parietal peritoneum. Thus, if the authors could revise that part once more so that the reason why such a clip must be installed could appear more clearly, this reviewer believes that their device concept will be more persuasive and attractive to other researchers. In addition, it was my overall feeling that the level of narrative ability for related contents still looks rudimentary, especially for the sections 2 and 3.
Answer 1 :
We have supplemented the content according to your comments. Please refer to lines of 48-64 (green) in the introduction and the underlined and green parts of 108-109 in session 2.
In session 2 and 3, I tried to supplement the content a little more. Please refer to lines of 136-139, 192-196, 214-215 (blue), and equations (7)-(8). After receiving a response to the accept decision, English sentences and contents will be continuously supplemented in the proofing version process to increase the degree of completeness. For reference, this study aims to quickly find a clip applied with neodymimum magnet using a detector, and compared the difference in performance and speed compared to other methods. Therefore, even if you feel regretful about circuit analysis or production related contents, we ask for your understanding with a generous heart. Thanks for the generous advice.
